# 10-Hydroxy-*trans*-2-decenoic Acid, a New Potential Feed Additive for Broiler Chickens to Improve Growth Performance

**DOI:** 10.3390/ani12141846

**Published:** 2022-07-20

**Authors:** Yuxin Zhang, Shixia Geng, Yuting Di, Yongbo Sun, Ying Liu, Juntao Li, Liying Zhang

**Affiliations:** State Key Laboratory of Animal Nutrition, Ministry of Agriculture and Rural Affairs Feed Industry Centre, China Agricultural University, Beijing 100193, China; s20203040632@cau.edu.cn (Y.Z.); gengshixia@163.com (S.G.); dyt971004@163.com (Y.D.); ybsun2014@163.com (Y.S.); liuyingcau2021@163.com (Y.L.); lijuntao@cau.edu.cn (J.L.)

**Keywords:** 10-hydroxy-*trans*-2-decenoic acid, broiler chicken, growth performance, immunity, antioxidant capacity

## Abstract

**Simple Summary:**

It is understood that 10-HDA is a medium-chain fatty acid derivative containing an ω-hydroxy group, which is one of the main active components in royal jelly. Furthermore, 10-HDA has been proven to possess various physiological and pharmacological activities, including antibiotic, immunomodulatory, and antioxidant activity, etc. However, few studies have shown the role of 10-HDA in broiler chickens. This research assessed the impacts of 10-HDA on growth performance, antioxidant capacity, and the immunity of broiler chickens. The findings demonstrated that a diet supplemented with 40 mg/kg 10-HDA apparently elevated the average daily gain, immunocompetence, and antioxidant capacity of broiler chickens. It is concluded that the growth performance of broiler chickens is greatly enhanced by 10-HDA, and that 10-HDA will have broad application prospects as a new feed additive in animal production.

**Abstract:**

The purposes of this study were to explore the potential possibility of 10-hydroxy-*trans*-2-decenoic acid (10-HDA) use in feeding broiler chickens. A total of 216 healthy 1-day-old chicks were divided into 2 treatments at random (diets supplemented with 0 or 40 mg/kg 10-HDA, respectively) with 6 replicates of 18 birds each, and were then reared for 42 days. The results found that a diet supplemented with 10-HDA significantly increased average daily gain of broiler chickens in d 22~42 and d 0~42. Compared with the control group, dietary inclusion of 10-HDA markedly increased the serum concentrations of immunoglobulin (Ig) G at d 21, as well as IgM and interleukin (IL)-10 at d 42, while decreasing the levels of tumor necrosis factor (TNF)-α at d 21, as well as IL-6, TNF-α, and IL-1β at d 42. Furthermore, broiler chickens fed a diet with 10-HDA had a higher (*p* < 0.05) serum activity of superoxide dismutase at d 42. Additionally, serum malondialdehyde content also decreased notably at d 21 and d 42. These results made it clear that 10-HDA increased the growth performance of broiler chickens, possibly by enhancing immune function and antioxidant capacity.

## 1. Introduction

Although 10-hydroxy-*trans*-2-decenoic acid (10-HDA) was first found in mandibular gland of worker bee in 1921, it was not extracted from royal jelly until 1957 by Butenandt and Rembold [1]. The molecular formula of 10-HDA is C_10_H_18_O_3_ (molecular weight, 186.25; CAS No.: 765-01-5) and the chemical structure is shown in Figure 1. Earlier studies have suggested that 10-HDA has broad-spectrum antibiotic activity [2] through damaging bacterial cell membrane structure, combining with bacterial DNA and further inhibiting bacterial DNA synthesis [3,4]. Later studies confirmed that immune function was enhanced by 10-HDA through regulating inflammatory pathways, reducing levels of proinflammatory cytokines [5,6], and promoting immunocyte activities [7]. In addition, 10-HDA also resisted oxidative stress by elevating the activity of antioxidizes and decreasing the production of reactive oxygen species [8]. Previously, 10-HDA was mainly obtained from natural royal jelly by physical extraction, which is an expensive process, so it was mainly applied in human healthcare products [9], cosmetics [10], and food [11] as a beneficial functional component. However, today 10-HDA is successfully synthesized by a chemical method in China with a relatively low price, which creates an opportunity for its application in animal production.

In the industrialized production of broiler chickens, broiler chickens are vulnerable to many stresses from technology, environment, and nutrition, etc. These stresses reduce the resistance to diseases and the performance of growing broiler chickens [12]. Improving the immunity and antioxidant capacity by scientific dietary formulation is a very effective measure to adapt to the actual production. The provider of 10-HDA used in this study conducted a preliminary experiment and found that providing 3.5 mg/kg body weight sodium 10-hydroxy-2-decenoate (10-HDANa) per day to broiler chickens via drinking water could increase the market body weight and survival rate (unpublished). Therefore, this study was designed to further probe into the impacts of diets supplemented with 10-HDA on the growth performance of broiler chickens using a standardized comparative experiment. We also aimed to explore if 10-HDA improved the immunity and antioxidant capacity, which provided evidence for the application of 10-HDA as a new feed additive in broiler chickens.

## 2. Materials and Methods

### 2.1. Experimental Material

Hebei Fengtong Biotechnology Co., Ltd. (Shijiazhuang, China) offered 10-HDA to this study, which is produced by chemical synthesis. The content of 10-HDA is over 97%, and the moisture and crude ash are less than 1.5%, respectively.

### 2.2. Experimental Design, Diets and Management

In total, 216 1-day-old healthy male chicks (Arbor Acres) were purchased from Arbor Acres Poultry Breeding Company (Beijing, China) and divided into 2 treatments (diets with 0 or 40 mg/kg 10-HDA) at random, with 6 replicates of 18 birds each for 42 days. The experiment was carried out in the poultry shelter of China Agricultural University (Beijing, China). The corn and soybean meal were two raw materials which were the basis in the formulation of the diets, and the diets met the nutritional requirements of the Nutrition Research Committee (1994) [13] for broiler chickens. The ingredient composition and nutrient concentrations are listed respectively in Table 1 and Table 2. During formulation of the diets, 10-HDA was firstly diluted 1000 times with corn flour by a step-by-step dilution, before being added together with other ingredients. The methods from the Association of Official Analytical Chemists (2000) [14] were used to analyze the amount of crude protein (method 988.05), calcium (method 927.02), lysine (method 994.12), and total phosphorus (method 995.11) in the diets. When methionine and cysteine were measured, acid hydrolysis with 6 M hydrochloric acid was performed after performic acid oxidation (method 994.12).

The broiler chickens were housed in three-decker wire-floored cages (0.9 × 0.6 × 0.4 m, six birds per cage) covered with clean plastic mats, and water and feed were given to them ad libitum. For the first three days, broiler chickens were provided 24 h light. Subsequently, the lights were turned off for 1 hour every day until d 42. During d 0~3, the temperature was kept at 34–35 °C, and then dropped slowly every week until the temperature reached 24–25 °C at d 35. This temperature was then maintained for the last week of the trial. The relative humidity was kept at 45–55%. All chicks were immunized on time with inactivated Newcastle disease vaccine on d 7 by eye-drop and nose-drop, and by drinking water on d 21 and, on d 14 and d 28, an infectious bursal disease vaccine (inactivated) was given via drinking water.

### 2.3. Growth Performance

The initial body weight of each broiler chicken was measured at the start of the trial. The weight of body and feed consumption per pen was noted after 8 h fasting at the end of the 21st and 42nd day, respectively. Additionally, average daily feed intake (ADFI), average daily gain (ADG), and the ratio of feed to gain (F:G) were calculated in d 0~21, d 22~42, and d 0~42, respectively. The formulas are as follows: ADG = average body weight gain per bird/days of rearing; ADFI = average feed consumption per bird/days of rearing; F:G = average feed consumption per bird/average body weight gain per bird.

### 2.4. Sample Collection and Storage

At the 21st and 42nd day of rearing, after 12 h fasting, six birds (a bird for each replicate) for each group that approached the mean of body weight were picked to take blood samples from the veins of wings into anticoagulant-free vacutainer tubes. The fresh blood samples collected were placed stably at room temperature for 30 min to coagulate. The blood samples were centrifuged at a speed of 3000× *g* for 15 min to obtain serums, and then serums were deposited at −20 °C for subsequent tests.

### 2.5. Sample Analyses

The contents of immunoglobulin (Ig) G, IgM, and IgA in serum were measured using an automatic biochemical analyzer (Hitachi 7600, Hitachi Group, Japan). The concentrations of IL-10, IL-6, IL-1β, and TNF-α in serum were measured using commercial kits (Beijing Leadman Biochemistry Co. Ltd., Beijing, China). The malondialdehyde (MDA) concentration, total antioxidant capacity (T-AOC), and the activity of glutathione peroxidase (GSH-Px) and superoxide dismutase (SOD) in serum were measured using commercial kits (Nanjing Jiancheng Institute of Bioengineering, Nanjing, China).

### 2.6. Statistical Analysis

An unpaired *t*-test procedure was utilized to analyze the data acquired from broiler chickens using SAS 9.2 (SAS Institute Inc., Cary, NC, USA). A significant difference between the two treatments was considered when *p* < 0.05.

## 3. Results

### 3.1. Growth Performance

The effects of 10-HDA on the growth performance of broiler chickens are listed in Table 3. Dietary supplementation of 10-HDA enhanced ADG (*p* < 0.05) at d 22~42 and d 0~42, but had no obvious impact on ADFI and F:G in each period (*p* > 0.05). During the whole trial, no diseases occurred for the broiler chickens in two groups, and the mortality was less than 2%.

### 3.2. Immunity

Dietary supplementation of 10-HDA markedly elevated IgG content and decreased the TNF-α content in serum at d 21. The concentrations of IL-10 and IgM notably increased. The levels of IL-1β, TNF-α, and IL-6 in serum at d 42 significantly reduced (Table 4). The levels of IgA, IgM, IL-1β, IL-6, and IL-10 in serum at d 21, as well as IgG and IgA at d 42, were not affected (*p* > 0.05).

### 3.3. Antioxidant Capacity

Dietary addition of 10-HDA increased SOD activity (*p* < 0.05) at d 42, while decreasing MDA content (*p* < 0.05) at d 21 and d 42 of the broiler chicken serum (Table 5). At d 21 and d 42, the difference in GSH-Px activity and T-AOC was not observed between two treatments (*p* > 0.05).

## 4. Discussion

The ADG, ADFI, and F:G are all important indicators to measure the economic benefit in animal production. In published studies, 10-HDA has been proved to be beneficial to the body weight gain of mice. Weiser et al. (2018) [15] reported that oral administration of 30 and 60 mg/kg BW 10-HDA each day for 120 days increased weight gain of male mice, and a notable rise in muscle mass and adipose tissue was attributed to the estrogenic effects of 10-HDA, which mediated estrogen signaling partly by modulating estrogen receptors. Fan et al. (2020) [7] also found that intragastrical administration of 0.1 g/kg BW 10-HDA per day for one week rescued weight loss of mice challenged with cyclophosphamide, and that there was a wide enhancement of DNA, RNA and protein behaviors induced by 10-HDA. However, none of the literature so far reported showed the influence of 10-HDA on the growth performance in broiler chickens. The actual study indicated that a diet addition of 40 mg/kg 10-HDA increased the growth performance of broiler chickens, although the optimal level in diet needs to be further explored.

Broiler chickens may be in a subhealth state in commercial intensive production, which increases morbidity and mortality and decreases growth performance [16]. Wang et al. [17] have proved that the improvement of immunity helps to reduce the impacts of subclinical diseases on growth performance. The serum levels of immunoglobulin including IgM, IgG, and IgA are the main indicators of the immune response of animals. Furthermore, TNF-α and IL-1β activate cytokine cascade in an inflammatory reaction in the early stage. The IL-10 represses the production of proinflammatory cytokines and enhances the function of cytotoxic T cells and B cells [18]. Otherwise, the IL-6 as a downstream mediator is produced by the stimulation of TNF-α and IL-1β [19]. Previous studies [5,6] have demonstrated that 10-HDA reduced inflammatory reaction by decreasing the production of upstream IFN-β or TNF-α, inhibiting the start of nuclear factor-kappa B (NF-κB) pathways and decreasing the production of downstream IL-6. Chen et al. (2018) [20] reported that oral administration of 0.1 g/kg BW 10-HDA per day for one week lessened the level of IL-10, IL-6, and TNF-α in serum of mice. Intragastrical administration with 0.1 g/kg BW 10-HDA per day for one week promoted the growth of immune organs, as well as revived the multiplication and function of T cell and B cell in the thymus and spleen [7]. The current study also made it clear that dietary inclusion of 40 mg/kg 10-HDA stimulated the secretion of immunoglobulin and anti-inflammatory cytokines, and showed that it reduced the level of proinflammatory cytokines. However, the effect of 10-HDA on serum IL-10 is different between our study and the research of Chen et al. (2018) [20]. We guessed that it was related to animals’ species and physical status. The inflammatory response caused by lipoteichoic acid in the research of Chen et al. was inhibited by 10-HDA in mice but, in our study, 10-HDA improved the anti-inflammatory ability of broiler chickens in non-stress conditions. Except for the suppression of NF-κB pathways, the inhibition of mitogen-activated protein kinase signaling pathways has also been proved to be one of the molecular mechanisms of 10-HDA in reducing inflammation [20]. Moreover, Šedivá and Klaudiny [3] thought that 10-HDA inhibited the pathogenic microbes by affecting the cell membrane, various basic processes related to the membrane, and the activity of enzymes. These antibiotic effects enhanced the ability of 10-HDA to exert immune response. Huang et al. (2022) [21] reported that 10-HDA relieved inflammation, and then reduced the weight loss in mice with colitis by enhancing the number of microvilli on the brush border, occludin, and zonula occludens-1 in the colon. Hence, we deduced that 10-HDA probably enhanced immunity through the above mechanisms, then improved intestinal health as well as digestion and absorption of nutrients, and finally promoted growth in broiler chickens.

Antioxidant capacity is a vital factor in influencing the health and performance of animals. The SOD is a crucial antioxidant enzyme that promotes the decomposition of superoxide anion into oxygen and into the less reactive hydrogen peroxide. The peroxides were catalyzed into water or alcohol by GSH-Px [22]. The T-AOC is used to comprehensively measure the antioxidant capacity of the body. The MDA is the ultimate peroxide of polyunsaturated fatty acids and other lipids attacked by free radicals, which reflects the degree of cell damage [23]. Therefore, MDA, SOD, T-AOC, and GSH-Px are usually chosen as markers of the status of antioxidant capacity in animals. A previous study found that 10-HDA and LPS co-treated macrophages significantly increased SOD activity and glutathione content, while decreasing the generation of free radicals including reactive oxygen species and nitric oxide, compared with an LPS-treated group [8]. In addition, oral administration of 2.5 and 5 mg/kg BW 10-HDA per day for 2 weeks in Ehrlich solid tumor mice strengthened the activity of SOD, catalase enzyme, and GSH-Px in tumor tissue, while lessening the concentrations of lipid peroxidation and nitric oxide [24]. This study verified that dietary inclusion of 40 mg/kg 10-HDA strengthened the antioxidant capacity of broiler chickens and alleviated oxidative damage. Huang et al. (2022) [21] recently found that 10-HDA enhanced the ability to resist oxidative stress by improving the expression of the thioredoxin/thioredoxin interacting protein ratio in mice. Combined with previous findings, we concluded that 10-HDA activated both the thioredoxin system and glutathione. As a transcription factor, nuclear factor erythroid-2 related factor 2 (Nrf2) is crucial to exert antioxidant protection and preserve redox homeostasis [25]. Furthermore, DNA methylation and histone modification are important epigenetic regulation methods, affecting Nrf2 signaling pathways [26]. Makino et al. (2016) [27] demonstrated that 10-HDA inhibited the activity of histone deacetylase to promote the expression of extracellular superoxide dismutase resulting from the enrichment of acetylated histone H4 in the proximal promoter region. Thus, it is reasonable to speculate that 10-HDA is a histone deacetylase that activates Nrf2, and then initiates the expression of antioxidant genes [28]. In addition, oxidative stress leads to many inflammations [29]. It was found that 10-HDA relieved inflammation induced by reactive oxygen species, increased immunity and antioxidation capacity, and then improved the health of intestine [21]. Therefore, we deduced that the growth-promoting effect of 10-HDA is also related to its antioxidant function. Therefore, 10-HDA protects lipids from oxidation, which prevents damage of cell membrane structures and enhances the utilization of nutrients in broiler chickens. As a result, the growth performance improved.

## 5. Conclusions

Dietary supplementation of 40 mg/kg 10-HDA improved the growth performance of broiler chickens, partly by elevating immunity and antioxidant capacity. This study reveals that the growth performance of broiler chickens is greatly enhanced by 10-HDA and, as a result, 10-HDA will have broad application prospects as a new feed additive in animal production.

## Figures and Tables

**Figure 1 animals-12-01846-f001:**
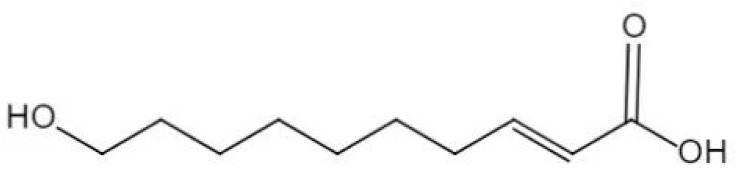
Chemical structure of 10-hydroxy-*trans*-2-decenoic acid.

**Table 1 animals-12-01846-t001:** Components and proportions of ingredients in the basal diet (%).

Ingredients	0~21 d	22~42 d
Corn	60.13	61.53
Soybean meal	32.50	31.70
Fish meal	2.00	0.00
Soybean oil	1.50	3.00
Dicalcium phosphate	1.50	1.70
Limestone	1.34	1.15
DL-methionine, 98%	0.23	0.12
Sodium Chloride	0.30	0.30
Premix ^1^	0.50	0.50
Total	100.00	100.00

^1^ The following are provided per kilogram diet: axerophthol, 9000 IU; cholecalciferol, 3000 IU; tocopherol, 24 mg; vitamin K_3_, 1.8 mg; thiamine, 2.0 mg; vitamin B_2_, 5.0 mg; nicotinic acid, 0.04 g; vitamin B_5_, 15 mg; vitamin B_6_, 3.0 mg; biotin, 50 µg; vitamin B_9_, 1.0 mg; vitamin B_12_, 100 µg; choline chloride, 0.5 g; iron (from ferrous sulfate monohydrate), 0.08 g; copper (from copper sulfate pentahydrate), 0.02 g; zinc (from zinc sulfate monohydrate), 0.09 mg; iodine (from KI), 350 µg; selenium (from Na_2_SeO_3_), 300 µg; manganese (from MnSO_4_·H _2_O), 0.08 g.

**Table 2 animals-12-01846-t002:** Nutrient concentrations of diets in two groups (%).

Nutrient Concentrations (%) ^1^	10-Hydroxy-*trans*-2-decenoic Acid (mg/kg)	
	0	40
0~21 d		
Metabolizable energy (MJ/kg)	12.59	12.59
Crude protein	21.51	21.54
Calcium	1.06	1.09
Total phosphorus	0.69	0.70
Lysine	1.20	1.18
Methionine + Cysteine	0.92	0.93
22~42 d		
Metabolizable energy (MJ/kg)	13.22	13.22
Crude protein	20.31	20.21
Calcium	0.92	0.91
Total phosphorus	0.66	0.67
Lysine	1.14	1.13
Methionine + Cysteine	0.85	0.85

^1^ Metabolizable energy is the theoretical value, and other data are analyzed values as described above.

**Table 3 animals-12-01846-t003:** Effects of 10-hydroxy-*trans*-2-decenoic acid on the growth performance of broiler chickens ^1^.

Item ^2^	10-Hydroxy-*trans*-2-decenoic Acid (mg/kg)		SEM	*p* Value
	0	40		
BW at d 42 (kg)	2.24	2.35	0.028	0.014
0~21 d				
ADG (g)	31.5	32.2	0.293	0.111
ADFI (g)	42.3	42.6	0.285	0.395
F:G	1.34	1.32	0.016	0.443
22~42 d				
ADG (g)	73.1	77.9	1.03	0.008
ADFI (g)	131	135	1.30	0.080
F:G	1.79	1.73	0.030	0.190
0~42 d				
ADG (g)	51.3	53.9	0.621	0.013
ADFI (g)	84.7	86.6	0.658	0.076
F:G	1.65	1.61	0.023	0.165

^1^ Values are the means of all birds for each treatment. ^2^ Here, ADFI = “average daily feed intake”; ADG = “average daily gain”; BW = “body weight”; F:G = “feed:gain ratio”.

**Table 4 animals-12-01846-t004:** Effects of 10-hydroxy-*trans*-2-decenoic acid on serum immune indicators of broiler chickens ^1^.

Item ^2^	10-Hydroxy-*trans*-2-decenoic Acid (mg/kg)		SEM	*p* Value
	0	40		
21 d				
IgG (g/L)	5.64	6.82	0.370	0.048
IgA (g/L)	0.915	0.883	0.041	0.589
IgM (g/L)	0.781	0.878	0.035	0.079
IL-1β (pg/mL)	43.2	39.1	2.57	0.287
TNF-α (pg/mL)	87	72.6	2.80	0.005
IL-6 (pg/mL)	72.4	69.2	1.71	0.206
IL-10 (pg/mL)	39.9	41.6	1.77	0.521
42 d				
IgG (g/L)	7.13	8.09	0.439	0.151
IgA (g/L)	1.24	1.19	0.060	0.557
IgM (g/L)	0.729	0.897	0.030	0.003
IL-1β (pg/mL)	42	28.5	1.16	<0.001
TNF-α (pg/mL)	83.1	65.8	2.09	<0.001
IL-6 (pg/mL)	75.1	62.8	1.41	<0.001
IL-10 (pg/mL)	35.9	43.7	1.42	0.003

^1^ Values are the average value of six birds picked for each treatment. ^2^ Here, IgA = “immunoglobulin A”; IgG = “immunoglobulin G”; IgM = “immunoglobulin M”; IL-10 = interleukin 10; IL-6 = “interleukin 6”; TNF-α = “tumor necrosis factor α”IL-1β = “interleukin 1β”.

**Table 5 animals-12-01846-t005:** Effects of 10-hydroxy-*trans*-2-decenoic acid on serum antioxidant indicators of broiler chickens ^1^.

Item ^2^	10-Hydroxy-*trans*-2-decenoic Acid (mg/kg)		SEM	*p* Value
	0	40		
21 d				
T-SOD (U/mL)	114	129	6.66	0.164
GSH-PX (U/mL)	352	358	9.16	0.632
T-AOC (mmol/L)	0.311	0.318	0.010	0.601
MDA (nmol/mL)	4.19	3.17	0.257	0.019
42 d				
T-SOD (U/mL)	134	153	1.62	< 0.001
GSH-PX (U/mL)	370	392	12.1	0.222
T-AOC (mmol/L)	0.323	0.318	0.008	0.649
MDA (nmol/mL)	3.39	2.32	0.154	0.001

^1^ Values are the average value of six birds picked for each treatment. ^2^ Here, MDA = “malonaldehyde”; T-AOC = “total antioxidant capacity”; GSH-Px = “glutathione peroxidase”; T-SOD = “total superoxide dismutase”.

## Data Availability

The corresponding author can provide the data presented in this study if necessary.

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
