# Peer review of "10-Hydroxy-trans-2-decenoic Acid, a New Potential Feed Additive for Broiler Chickens to Improve Growth Performance"

_animals, 2022, doi:10.3390/ani12141846_

Round 1

Reviewer 1 Report

In the chapter 2.2. Experimental design, diets and management is stated that synthetic 10- hydroxy-trans-2-decenoic acid was administered to chickens diet (feed mixture) in the amount of 40 mg / kg. How was such a small amount (0.004%) evenly mixed into the diets?

In the Table 1 (composition of the diets), the 10-HDA is not included as part of the premix or as separate component. Could you explain it?   The authors declare that 10-DHA is a suitable nutritional supplement in poultry nutrition, it can improve the growth performance by enhancing immunity and antioxidant capacity. Therefore, in connection with this, it is necessary to describe the health status of broilers and mortality in the control and experimental groups.  

The part of Discussion is more of a literature review as a real discussion. The authors reported the effects of 10-HDA from several published experiments with rodents. I recommend also the study of Šedivá and Klaudiny  - The antimicrobial substances of Royal Jelly, In Chem. Letters, 109, 755-761 (2015).

Reviewer 2 Report

Dear Authors,

- The whole manuscript needs to be revised with a native speaker/editor. There are many statements that can be improved or there are grammatical errors.  Please check the whole manuscript. For example Line 52:  For industry production of broiler chickens, it is facing many stresses which could reduce the resistance to diseases and have negative effects on growth performance. 

- Line 10. It requires to briefly introduce 10-hydroxy-trans-2-decenoic acid.

- Line 12. Please rewrite the following statement.  But it is not found any study on the application of 10-HDA in animal 12 production until now

Line 16 and 17. The conclusion is no strong for a novel study. Please rewrite it and combine two statements. This is the first report on the application of  10-HDA in broiler chickens, and we think 10-HDA has the possibility to be applied as a new feed additive in broiler chickens.

- Line 46-51. This part should combine with the first paragraph. Possibly before line 37.  

-Line 66. Why selected 40mg/kg not higher or lower? Please add your response to the main text and cite it.

- Please add more details about the experiment such RH, cage/pen size and bed condition. Furthermore, please provide more information about the additive as a table. This is a new additive in the field and reader would be interested to see what are the contents.  

-Line 95 and 96 section 2.3. Please rewrite it. 

- There is no need to have superscripts when there are two groups only and P values are available. Please remove them all. Also, remove the statements under tables regarding superscripts. 

-Table 3. Please add the final body weight at day 42. Please re-analysis the data for ADG days 22-42 and 0-42 and confirm the current analysis.

- Table 5. Level of MDA is high for both groups. Did they birds experienced any stress condition?

Regards, 

Reviewer 3 Report

General comments: the objective of the reported experiment is to study growth performance effect of an additive (10-HDA) and the impact on immunity. Two diets were compared. The study was short and simple and not complicated in design. However, the writing is poor in many places.  part of the manuscript reads well whereas the many other parts are not properly constructed, with poor grammar, writing and similar issues. The authors will need thorough English editing for the entire manuscript. The authors also need to do more than stating what was observed (in the discussion section) or simply comparing with what others found. A good treatise of the results is expected, where the authors fully explain implication of their findings. It is important for the authors to try to account for the link between the immune aspects and growth performance and put this in context. The authors have not done a good job of accounting for growth performance and this is a main weakness in the manuscript. For example, in the absence of any immune challenge, what is the relevance of the immune aspects measured? Other comments are also below.

L12-3: please rewrite this sentence

L23: is it could increase, or increased…?

L25: statement on IgM should be checked, p value is 0.08, higher than 0.05 that is significant…

L25: statement regarding IL10 is not correct

L45: functional?

L46-51: this doesn’t  read well. please rewrite.

L71: Tables

L78: for the first three days…

L81: at what age did temp reach 24C?

L84: which vaccine was by eye drop, which one is by nose drop and which one is by water? Please be specific

Table 1: is it necessary to report corn and SBM level to 2 decimal places. Your scales can not be that accurate

L108; 112: measured by commercial kits…

Table 3: please use decimal places judiciously, e.g. 131 needs no decimal places, values with two full numbers need only one decimal place etc. P values should be three decimal places

L128: “could” can not be used in results. The results should state what was observed not what could be observed…

L142: please rewrite this sentence because it doesn’t read well

L154: could increase, or increased?

L155: rise or increase, not raise

L157, 159: see previous comment

L167: please rewrite sentence

L191: chosen

L194: reactive oxygen species…

L207-10: the conclusion should be rewritten so it is not sounding like possibility (could) but more definite (e.g. improved growth performance by ….)

Round 2

Reviewer 2 Report

Dear Authors, 

Thank you for the revised version. I have no more comments.

Regards,